# Learning to dehaze with polarization

**Chu Zhou**[1] **Minggui Teng**[2] **Yufei Han**[5] **Chao Xu**[1] **Boxin Shi**[2,3,4*]
[1]Key Lab of Machine Perception (MOE), Dept. of Machine Intelligence, Peking University
[2]Nat'l Eng. Lab for Video Technology, School of Computer Science, Peking University
[3]Institute for Artificial Intelligence, Peking University
[4]Beijing Academy of Artificial Intelligence
[5]School of Info. and Comm. Eng., Beijing University of Posts and Telecommunications
{zhou_chu, minggui_teng, shiboxin}@pku.edu.cn,
hanyufei@bupt.edu.cn, xuchao@cis.pku.edu.cn

## Abstract

Haze, a common kind of bad weather caused by atmospheric scattering, decreases the visibility of scenes and degenerates the performance of computer vision algorithms. Single-image dehazing methods have shown their effectiveness in a large variety of scenes, however, they are based on handcrafted priors or learned features, which do not generalize well to real-world images. Polarization information can be used to relieve its ill-posedness, however, real-world images are still challenging since existing polarization-based methods usually assume that the transmitted light is not significantly polarized, and they require specific clues to estimate necessary physical parameters. In this paper, we propose a generalized physical formation model of hazy images and a robust polarization-based dehazing pipeline without the above assumption or requirement, along with a neural network tailored to the pipeline. Experimental results show that our approach achieves state-of-the-art performance on both synthetic data and real-world hazy images.

## 1 Introduction

When taking photos in hazy environments, the visibility and color fidelity of recorded scenes are usually contaminated, because the captured images often contain a superposition of two unknown components: the transmitted light (an attenuated fraction of original scene radiance), and the airlight (ambient light scattered towards the viewer). It is highly ill-posed to separate them in a single hazy image as it requires estimating multiple unknowns from a single observation. Handcrafted priors from natural image statistics [21, 15, 1] have been wildly used to solve this problem. With the development of deep neural networks, learning-based methods (*e.g.*, CNN-based [24, 65, 7, 82] and GAN-based [79, 5]) have also been adopted to recover the haze-free images by extracting image features from a large amount of training data. However, these methods do not generalize well to real-world images, because they depend strongly on the image features extracted from training data and do not explicitly consider useful constraints from physical image formation models.

For better generalization, multi-image dehazing methods have been proposed. They capture multiple images from different viewpoints [38, 52, 84, 63], weather conditions [58, 55, 56, 57], or polarization angles [77, 78, 53, 81, 76, 25, 54]. Although all of these multi-image dehazing methods can relieve the ill-posedness, polarization-based ones have their unique advantages, since they directly utilize the physical image formation model with less dependency on image features extracted from training data. Nowadays, multiple polarized images can be conveniently captured in a single shot using a

---

*Corresponding author.

35th Conference on Neural Information Processing Systems (NeurIPS 2021).

polarization camera such as Lucid Vision Phoenix polarization camera[2]. However, these polarization-based methods are not robust due to several issues:

(1). They are largely based on a strong assumption that the transmitted light is not significantly polarized, while it is not the case for real-world images since both transmitted light and airlight contribute to the polarization [13, 29].

(2). They usually require specific clues (*e.g.*, sky regions [13, 77, 53], similar objects [54, 50], known depth [54]) to estimate the infinite airlight and the degree of polarization (DoP), which significantly reduces their applicability since these requirements are not always met.

(3). They are optimization-based methods which do not make full use of semantic and contextual information in image features to handle the spatially-variant real-world scattering.

In this paper, to enable the polarization-based dehazing methods to handle images captured in the wild more robustly, we propose a generalized physical formation model of hazy images, without assuming that the transmitted light is not significantly polarized, while considering the spatially-variant real-world scattering. Based on the physical model, we propose a robust polarization-based dehazing pipeline to extend their applicability by adopting deep learning to estimate the infinite airlight and the DoP of both transmitted light and airlight without the requirement of specific clues like sky regions, similar objects, *etc*. According to our dehazing pipeline, we design a neural network to perform the dehazing process: It first estimates the DoP of both transmitted light and airlight to solve the transmitted light, then predicts the infinite airlight to reconstruct the original scene radiance. Thanks to our learning-based pipeline, our method extracts image features from training data and use semantic and contextual information to refine the results, which is suitable for handling the spatially-variant real-world scattering.

To summarize, this paper makes contributions by demonstrating:

(1). A generalized physical formation model of hazy images, taking into account the polarization effects of both transmitted light and airlight, along with the spatially-variant real-world scattering.

(2). A robust polarization-based dehazing pipeline without the requirement of specific clues, by adopting deep learning to estimate necessary physical parameters (infinite airlight, DoP of both transmitted light and airlight).

(3). A neural network making full use of semantic and contextual information to handle the spatially-variant real-world scattering to improve the clarity of original scene radiance recovery.

Experimental results show that our approach achieves state-of-the-art performance on both synthetic data and real-world hazy images.

## 2  Related work

**Single-image dehazing.** Single-image dehazing is a highly ill-posed problem because it requires estimating multiple unknowns (the transmitted light and airlight) from a single observation. Park *et al*. [64] estimated haze from the difference among the RGB channels. Some methods adopted an adaptive contrast enhancement strategy to maximize the local contrast of restored images [87, 17, 47]. Some works proposed several assumptions (*e.g.*, the surface shading and transmission are locally uncorrelated [14], the scene albedo and depth are independent [59], both the scene albedo and transmission are constant inside each patch [86]) or image priors (*e.g.*, dark channel prior [21, 89, 49, 37], color attenuation prior [98], non-local prior [1, 43], ellipsoid prior [19], color-lines [15]) to handle this problem. Recently, with the development of deep neural networks, learning-based methods have also been adopted to recover haze-free images by extracting image features from a large amount of training data. These learning-based methods could be divided into two groups: direct methods, which dehaze in an end-to-end manner using convolutional neural networks (CNN) [2, 68, 30, 31, 62, 6, 18, 45, 95, 44, 36, 24, 65, 7, 8, 70, 94, 51, 4, 48, 3] or generative adversarial networks (GAN) [93, 69, 35, 92, 66, 33, 34, 79, 5, 9, 12, 85, 10, 82], and indirect methods, which

---

[2]https://thinklucid.com/product/phoenix-5-0-mp-polarized-model/

estimate image priors or their variants [88, 74, 91, 46, 20] first and then use them to dehaze. Although these methods have shown their effectiveness in a large variety of scenes, their generalization ability is still limited, since image priors are not always observed in the input and the image features extracted from synthetic training data often have a large domain gap with real-world ones.

**Multi-image dehazing.** For better generalization and less ill-posedness, multi-image dehazing methods have been proposed. They use computational photography techniques to capture multiple images with conventional or unconventional cameras for acquiring extra information. Some methods used separately measured range data [60] and georeferenced digital terrain with urban models [28] to facilitate dehazing. Some works captured multiple images under different unknown weather conditions to recover scene depth maps for dehazing [58, 55, 56, 57]. Some approaches take advantage of stereo vision to remove the effects of haze [38, 52, 84, 63] by taking multiple photos from different viewpoints. Some methods fuse RGB images with NIR (near-infrared) ones to help dehaze [75, 16, 42, 11, 83], because the scattering is significantly smaller in NIR than in visible light since NIR wavelengths are longer. Although these methods have better generalization ability, capturing such data is not easy since they require multiple shots and/or complicated imaging systems.

**Polarization-based dehazing.** Recently, polarization-based methods have been proposed to solve the dehazing problem by capturing multiple polarized images at the same view with different polarization angles. These methods have their unique advantages: they directly utilize the physical image formation model without dependence on image features extracted from training data, and multiple polarized images can be captured in a single shot using a polarization camera. However, most of them are based on a strong assumption that the transmitted light is not significantly polarized [77, 78, 53, 81, 76, 25, 54, 50, 41, 40, 39, 67, 80], while it is not the case for real-world images since both transmitted light and airlight contribute to the polarization [13, 29]. Fang *et al*. [13] takes the polarization of transmitted light into consideration, however, it supposes that the depth of sky regions is approximately infinite and require sky regions to estimate the infinite airlight and the DoP, just like [77, 53], which significantly reduces the applicability since sky regions are not always available.

## 3 Method

In this section, we show the physical formation model of hazy images in Section 3.1, demonstrate our polarization-based dehazing pipeline in Section 3.2, and introduce our neural network in Section 3.3.

### 3.1 Physical image formation model

As shown in Figure 1 (top row), when taking photos in hazy environments, caused by atmospheric scattering, the captured image $\mathbf{I} = \{I(x, y, c)\}$ ($(x, y)$ is the pixel coordinate and $c$ denotes the color channel index) is composed of two components: the transmitted light $\mathbf{T} = \{T(x, y, c)\}$ (an attenuated fraction of original scene radiance $\mathbf{R} = \{R(x, y, c)\}$), which decreases with the scene depth $\mathbf{z} = \{z(x, y)\}$, and the airlight $\mathbf{A} = \{A(x, y, c)\}$ (ambient light scattered towards the viewer), which increases with $\mathbf{z}$. According to [77], the formation of a hazy image can be described as

$$\mathbf{I} = \mathbf{T} + \mathbf{A} = \mathbf{R} \cdot e^{-\boldsymbol{\beta} \cdot \mathbf{z}} + \mathbf{A}_\infty \cdot (1 - e^{-\boldsymbol{\beta} \cdot \mathbf{z}}), \tag{1}$$

where $\boldsymbol{\beta} = \{\beta(c)\}$ is the scattering coefficient, $\mathbf{A}_\infty = \{A_\infty(c)\}$ denotes the infinite airlight (the airlight radiance corresponding to an object at an infinite distance, *e.g.*, the horizon), and $\cdot$ stands for element-wise multiplication. A synthetic example of its visualization can be found in Figure 1 (bottom left). However, real-word scattering does not always satisfy such an ideal model, which means that both $\boldsymbol{\beta}$ and $\mathbf{A}_\infty$ are not only dependent on wavelength, but also on the the size of the scattering particles [27, 22] and angular scattering coefficient [78]. To encode such variations, we replace them with $\boldsymbol{\beta} = \{\bar{\beta}(c) + N(x, y, c)\}$ and $\mathbf{A}_\infty = \{\bar{A}_\infty(c) + N(x, y, c)\}$ respectively, where $\bar{\cdot}$ marks the mean value and $N(x, y, c)$ denotes the spatially-variant turbulence. Assume for a moment that the illumination of any scattering particle comes from one direction, the light ray from the source to a scatterer and the line-of-sight from the camera to the scatterer define a plane of incidence (PoI) [77], as shown in Figure 1 (top row). We decompose $\mathbf{I}$, $\mathbf{T}$, and $\mathbf{A}$ into two components respectively: $\mathbf{I}^\parallel$ and $\mathbf{I}^\perp$, $\mathbf{T}^\parallel$ and $\mathbf{T}^\perp$, $\mathbf{A}^\parallel$ and $\mathbf{A}^\perp$, where the subscript $\parallel$ ($\perp$) means the component is parallel (perpendicular) to the PoI. The degrees of polarization (DoP) of $\mathbf{I}$, $\mathbf{T}$, and $\mathbf{A}$ are defined as

$$\mathbf{P} \triangleq \frac{\mathbf{I}^\perp - \mathbf{I}^\parallel}{\mathbf{I}}, \quad \mathbf{P}_T \triangleq \frac{\mathbf{T}^\perp - \mathbf{T}^\parallel}{\mathbf{T}}, \quad \text{and} \quad \mathbf{P}_A \triangleq \frac{\mathbf{A}^\perp - \mathbf{A}^\parallel}{\mathbf{A}}, \tag{2}$$

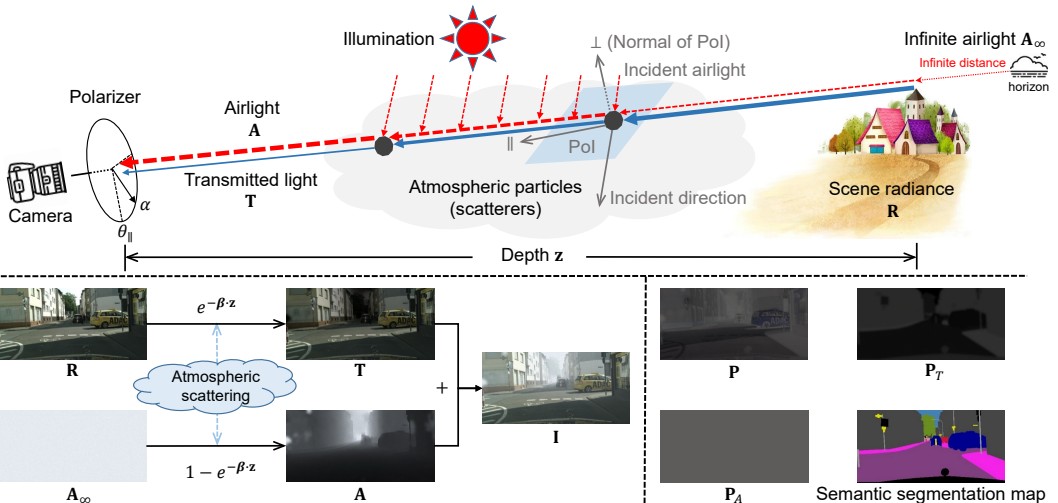

Figure 1: Top row: An illustration of the atmospheric scattering and polarization; transmitted light (blue solid line) **T** is an attenuated fraction of original scene radiance **R** that decreases with the scene depth **z**; airlight **A** (red dashed line) is the ambient light scattered towards the viewer that increases with **z**; when placing a linear polarizer with polarization angle $\alpha$ in front of the camera, the polarization component parallel to the plane of incidence (PoI) is best transmitted through the polarizer at $\alpha = \theta_\parallel$. Bottom left: A synthetic example for visualizing the formation of a hazy image (see Equation (1) for details). Bottom right: A synthetic example for visualizing $\mathbf{P}$, $\mathbf{P}_T$, and $\mathbf{P}_A$ (the DoP of **I**, **T**, and **A**, see Equation (2) for details), along with the semantic segmentation map.

respectively, where

$$\mathbf{I} = \mathbf{I}^\perp + \mathbf{I}^\parallel, \quad \mathbf{T} = \mathbf{T}^\perp + \mathbf{T}^\parallel, \quad \text{and} \quad \mathbf{A} = \mathbf{A}^\perp + \mathbf{A}^\parallel. \tag{3}$$

Since the scattered light is partially polarized perpendicular to the PoI [77, 27, 22, 13], $\mathbf{P}$, $\mathbf{P}_T$, and $\mathbf{P}_A$ are not less than zero. Besides, although $\mathbf{P}$ is spatially-variant (*i.e.*, $\mathbf{P} = \{P(x, y, c)\}$), the distributions of $\mathbf{P}_T$ and $\mathbf{P}_A$ are not irregular: the values of $\mathbf{P}_T$ are approximately uniform in the same semantic segment[3], while $\mathbf{P}_A$ can be regarded as spatially-uniform, *i.e.*, $\mathbf{P}_A = \{P_A(c)\}$, according to [13]. A synthetic example of their visualization can be found in Figure 1 (bottom right).

When we place a polarizer with polarization angle $\alpha$ in front of the camera, according to Malus' law [22], the captured polarized image $\mathbf{I}_\alpha$ can be calculated as

$$\mathbf{I}_\alpha = \frac{\mathbf{I} \cdot (1 - \mathbf{P} \cdot \cos(2(\alpha - \boldsymbol{\theta}_\parallel)))}{2}, \tag{4}$$

where $\boldsymbol{\theta}_\parallel = \{\bar{\theta}_\parallel + N(x, y, c)\}$ denotes the orientation of the polarizer for best transmission of the component parallel to the PoI. Similarly, the two components **T** and **A** at angle alpha can be calculated as

$$\mathbf{T}_\alpha = \frac{\mathbf{T} \cdot (1 - \mathbf{P}_T \cdot \cos(2(\alpha - \boldsymbol{\theta}_\parallel)))}{2} \quad \text{and} \quad \mathbf{A}_\alpha = \frac{\mathbf{A} \cdot (1 - \mathbf{P}_A \cdot \cos(2(\alpha - \boldsymbol{\theta}_\parallel)))}{2}, \tag{5}$$

which satisfy $\mathbf{I}_\alpha = \mathbf{T}_\alpha + \mathbf{A}_\alpha$. Note that both **T** and **A** contribute to the polarization, and the polarization of **T** should not be ignored [13, 29]. From Equation (4) and Equation (5), we can derive the following equation:

$$\mathbf{I} \cdot \mathbf{P} = \mathbf{T} \cdot \mathbf{P}_T + \mathbf{A} \cdot \mathbf{P}_A, \tag{6}$$

which reveals that the relationship among **I**, **T**, and **A** are determined by $\mathbf{P}$, $\mathbf{P}_T$, and $\mathbf{P}_A$.

---

[3]The polarization properties of transmitted light depend on material properties of scene objects (*e.g.*, surface texture) [29], and objects in the same semantic segment often have similar material properties.

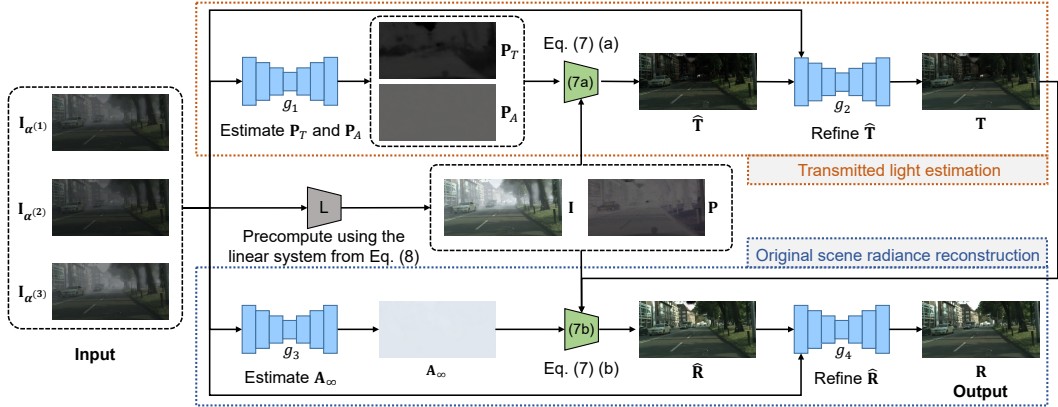

Figure 2: We design a network tailored to our polarization-based dehazing pipeline (in Section 3.2), which takes three polarized images $\mathbf{I}_{\alpha^{(i)}}(i = 1, 2, 3)$ captured at the same view with different polarization angles $\alpha^{(i)}(i = 1, 2, 3)$ as the input (along with the precomputed hazy image $\mathbf{I}$ and its DoP $\mathbf{P}$ using the linear system from Equation (8)) and outputs the reconstructed original scene radiance $\mathbf{R}$. It consists of two stages: transmitted light estimation and original scene radiance reconstruction. The first stage includes two subnetworks for estimating $\mathbf{P}_T$, $\mathbf{P}_A$, and refining $\widehat{\mathbf{T}}$. The second stage also includes two subnetworks for estimating $\mathbf{A}_\infty$ and refining $\widehat{\mathbf{R}}$. ($\widehat{\phantom{.}}$ denotes the coarse value calculated from Equation (7)).

## 3.2 Polarization-based dehazing pipeline

We aim to restore the original scene radiance $\mathbf{R}$ using three polarized images $\mathbf{I}_{\alpha^{(i)}}(i = 1, 2, 3)$ captured at the same view with different polarization angles $\alpha^{(i)}(i = 1, 2, 3)$. Eliminating $\mathbf{A}$ from Equation (1) and Equation (6), $\mathbf{T}$ and $\mathbf{R}$ could be computed by the following two equations:

$$\text{(a)} \quad \mathbf{T} = \frac{\mathbf{P} \cdot \mathbf{I} - \mathbf{I} \cdot \mathbf{P}_A}{\mathbf{P}_T - \mathbf{P}_A} \quad \text{and} \quad \text{(b)} \quad \mathbf{R} = \frac{\mathbf{T} \cdot \mathbf{A}_\infty}{\mathbf{A}_\infty - (\mathbf{I} - \mathbf{T})}, \tag{7}$$

where $\mathbf{P}_T$, $\mathbf{P}_A$, and $\mathbf{A}_\infty$ are required to be estimated, $\mathbf{I}$ and $\mathbf{P}$ can be directly calculated by $\mathbf{I}_{\alpha^{(i)}}(i = 1, 2, 3)$.

Now we first explain how to calculate $\mathbf{I}$ and $\mathbf{P}$ using $\mathbf{I}_{\alpha^{(i)}}(i = 1, 2, 3)$. Expanding Equation (4), we obtain ($\langle\rangle$ denotes inner product)

$$\mathbf{I}_\alpha = \left\langle \left[ \frac{1}{2} \quad \frac{-\cos(2\alpha)}{2} \quad \frac{-\sin(2\alpha)}{2} \right], [\mathbf{D}_1 \quad \mathbf{D}_2 \quad \mathbf{D}_3] \right\rangle,$$
$$\text{where} \quad \mathbf{D}_1 = \mathbf{I}, \quad \mathbf{D}_2 = \mathbf{I} \cdot \mathbf{P} \cdot \cos(2\boldsymbol{\theta}_\|) \quad \text{and} \quad \mathbf{D}_3 = \mathbf{I} \cdot \mathbf{P} \cdot \sin(2\boldsymbol{\theta}_\|). \tag{8}$$

Since Equation (8) has three unknowns ($\mathbf{D}_i(i = 1, 2, 3)$), we can use $\mathbf{I}_{\alpha^{(i)}}(i = 1, 2, 3)$ to obtain a linear system that allows to compute them. Then, we could calculate $\mathbf{I}$ and $\mathbf{P}$ by $\mathbf{I} = \mathbf{D}_1$ and $\mathbf{P} = \frac{\sqrt{(\mathbf{D}_2^2 + \mathbf{D}_3^2)}}{\mathbf{D}_1}$ respectively.

Next, we only need to estimate three parameters $\mathbf{P}_T$, $\mathbf{P}_A$, and $\mathbf{A}_\infty$ to reconstruct $\mathbf{R}$. To alleviate the dependency on specific clues (such as sky regions [13, 77, 53] or similar objects [54, 50], which are required by other polarization-based methods) for estimating these parameters, we choose to design a deep neural network that comprehensively explores physics and semantic features.

## 3.3 Polarization-based dehazing network

As shown in Figure 2, our network consists of two stages: transmitted light estimation and original scene radiance reconstruction.

**Transmitted light estimation.** As shown in the first stage of Figure 2, it aims to estimate the DoP of both transmitted light and airlight ($\mathbf{P}_T$ and $\mathbf{P}_A$) for solving the transmitted light $\mathbf{T}$. So, it adopts a subnetwork $g_1$ to estimate $\mathbf{P}_T$ and $\mathbf{P}_A$, then uses Equation (7) (a) to calculate $\widehat{\mathbf{T}}$ (the coarse value

of $\mathbf{T}$). However, we cannot directly feed $\widehat{\mathbf{T}}$ into the second stage since the numerical problem will occur when the denominator of Equation (7) (a) approaches zero, which often happens in pixels where $\mathbf{P}_T \approx \mathbf{P}_A$ (the DoP of transmitted light and airlight are approximately the same). Besides, the estimated $\mathbf{P}_T$ and $\mathbf{P}_A$ by $g_1$ are prone to be noisy which distorts the calculated $\widehat{\mathbf{T}}$, because the spatially-variant turbulence is hard to learn due to its irregularities. So, we adopt another subnetwork $g_2$ to refine $\widehat{\mathbf{T}}$ using semantic and contextual information extracted from $\mathbf{I}_{\alpha^{(i)}} (i = 1, 2, 3)$. In practice, we construct $g_1$ using the U-Net architecture [71] since it works well on per-pixel estimation tasks such as semantic segmentation [71, 61]. As for $g_2$, we choose the autoencoder architecture [23], by virtue of its excellent context generalization ability for refining image contents.

**Original scene radiance reconstruction.** As shown in the second stage of Figure 2, it aims to estimate the infinite airlight $\mathbf{A}_\infty$ to reconstruct the original scene radiance $\mathbf{R}$. So, it first adopts a subnetwork $g_3$ to estimate $\mathbf{A}_\infty$, then uses Equation (7) (b) to calculate $\widehat{\mathbf{R}}$ (the coarse value of $\mathbf{R}$). However, $\widehat{\mathbf{R}}$ also needs to be refined, because when the haze in some pixels is very thick and leaves little information of the transmitted light ($\mathbf{T} \approx \mathbf{0}$), the numerator of Equation (7) (b) approaches zero, which leads to a wrong result that $\mathbf{R} \approx \mathbf{0}$. So, similar to the first stage, we also adopt a subnetwork $g_4$ to refine $\widehat{\mathbf{R}}$. We also choose the U-Net architecture [71] for $g_3$ and the autoencoder architecture [23] for $g_4$.

# 4 Data preparation and network training

In this section, we first detail our synthetic dataset generation pipeline in Section 4.1, then show our loss function and training strategy in Section 4.2.

## 4.1 Synthetic dataset generation pipeline

It is difficult to obtain pairwise hazy and clear images with three polarized observations at a large scale. Besides, getting the ground truth values of the DoP or infinite airlight is not feasible. So, we propose to generate a synthetic dataset for training our network. Since we require spatially-variant $\boldsymbol{\beta}$ and $\mathbf{A}_\infty$ to simulate real-world scattering, and need the semantic segmentation map $\mathbf{S}$ for generating reasonable $\mathbf{P}_T$ (see Section 3.1 for details about the properties of $\mathbf{P}_T$), we cannot directly generate the polarized images from the hazy images in existing dehazing benchmarks [32, 97, 96, 73, 72]. The desired data source for generating our dataset should provide:

(1). clear image $\mathbf{R}$ with depth map $\mathbf{z}$, from which we can calculate $\mathbf{I}$ using Equation (1) by generating spatially-variant $\boldsymbol{\beta}$ and $\mathbf{A}_\infty$;

(2). semantic segmentation map $\mathbf{S}$, from which we can generate reasonable $\mathbf{P}_T$ using $\mathbf{P}_T = f(\mathbf{S})$, where $f$ denotes a function which randomly maps each semantic segment to a value of $\mathbf{P}_T$.

The Foggy Cityscapes-DBF dataset [72] meets the above two requirements[4], so we use the provided $\mathbf{z}$, $\mathbf{R}$, and $\mathbf{S}$ to generate our synthetic dataset. In short, with $\mathbf{z}$, $\mathbf{R}$, and $\mathbf{S}$ available, our synthetic dataset generation pipeline could be described as[5]:

(1). Randomly generate $\boldsymbol{\beta}$ (in $[0.01, 0.02]$), $\mathbf{A}_\infty$ (in $[0.85, 0.95]$), and $\mathbf{P}_A$ (in $[0.05, 0.4]$) to calculate $\mathbf{T}$, $\mathbf{A}$, and $\mathbf{I}$ using Equation (1);

(2). generate $\mathbf{P}_T$ from $\mathbf{S}$ using $\mathbf{P}_T = f(\mathbf{S})$ (in $[0.025, 0.2]$), then calculate $\mathbf{P}$ using Equation (6);

(3). randomly generate $\boldsymbol{\theta}_\parallel$ (in $[-45°, 45°]$), then use Equation (4) to calculate $\mathbf{I}_{\alpha^{(i)}} (i = 1, 2, 3)$ ($\alpha^{(i)} (i = 1, 2, 3)$ are set to be $0°$, $45°$, and $90°$ respectively).

The visualization of above mentioned parameters can be found in the bottom row of Figure 1. Note that for $\boldsymbol{\beta} = \{\bar{\beta}(c) + N(x, y, c)\}$ and $\mathbf{A}_\infty = \{\bar{A}_\infty(c) + N(x, y, c)\}$, we first generate their mean values $\bar{\beta}(c)$ and $\bar{A}_\infty(c)$ for each channel, then add 5% Gaussian noise to make them spatially-variant. Besides, we also add 2% Gaussian noise to $\mathbf{I}_{\alpha^{(i)}} (i = 1, 2, 3)$. To conform to the real-world scattering

---

[4]Although it does not directly provide $\mathbf{z}$, it offers the transmittance ($e^{-\boldsymbol{\beta} \cdot \mathbf{z}}$) with known spatially-uniform scattering coefficient $\boldsymbol{\beta}$, so that we can can compute $\mathbf{z}$ by ourselves.

[5]The range of $\boldsymbol{\beta}$ is from Li *et al.* [32] with some adjustment (changing the sampling space from discrete to continuous), and the ranges of $\mathbf{P}_A$ and $\mathbf{P}_T$ are from the statistics in Fang *et al.* [13].

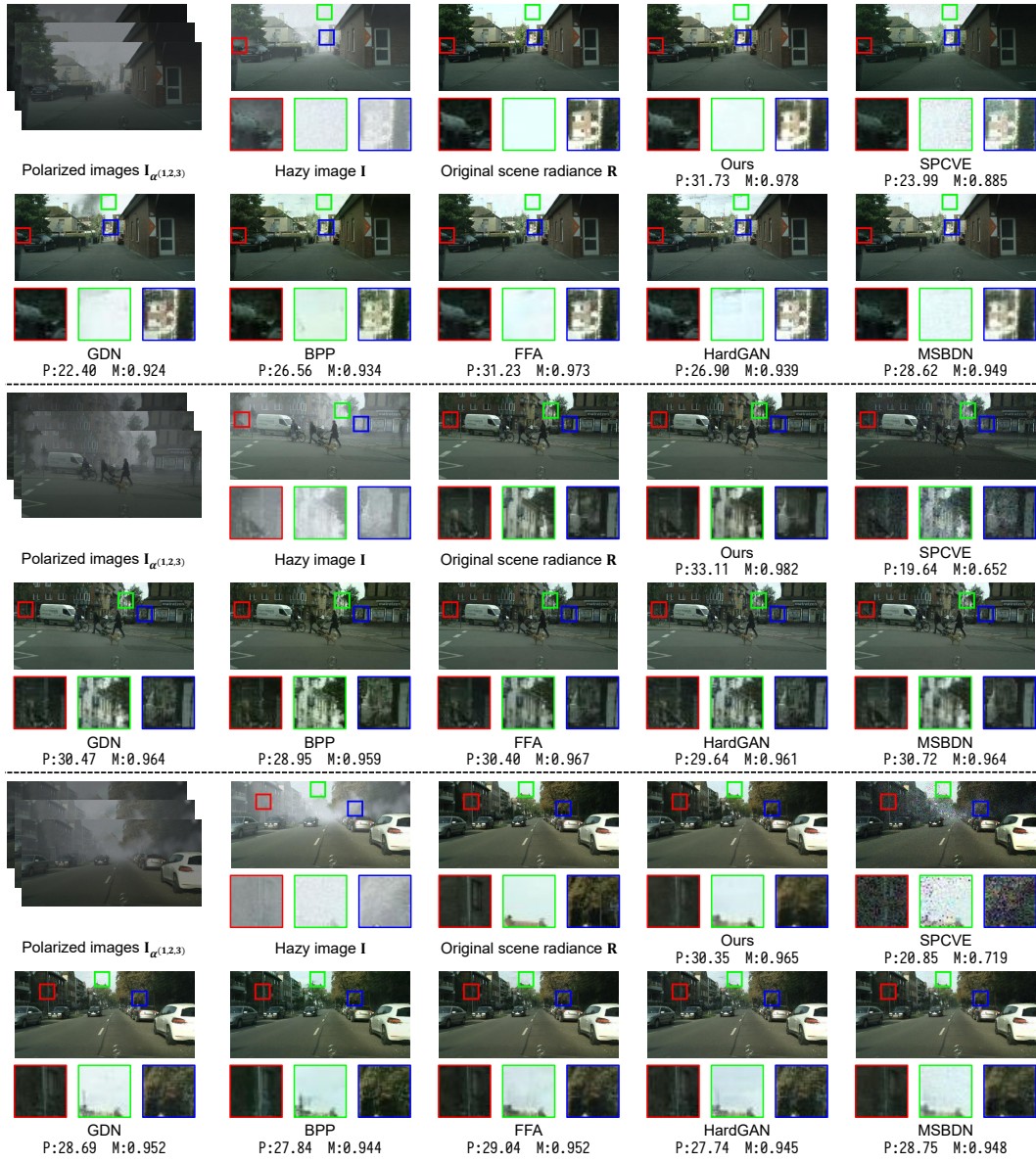

Figure 3: Qualitative comparisons on synthetic data among our method, a representative polarization-based dehazing algorithm SPCVE [54] which also takes three polarized images as the input, and five state-of-the-art learning-based dehazing methods including GDN [44], BPP [82], FFA [65], HardGAN [5], and MSBDN [7] which take a single hazy image as the input. Quantitative results evaluated using PSNR (P) and MS-SSIM (M) are displayed below each image.

[77], we ensure that $\bar{\beta}(r) < \bar{\beta}(g) < \bar{\beta}(b)$ and $P_A(r) > P_A(g) > P_A(b)$. The images are resized and randomly cropped to $240 \times 240$ patches during the training process, and cropped to $496 \times 240$ patches for test[6].

## 4.2   Loss function and training strategy

**Loss function.** The total loss function of our network is $\mathcal{L} = \lambda_1 \cdot \mathcal{L}_{g_1} + \lambda_2 \cdot \mathcal{L}_{g_2} + \lambda_3 \cdot \mathcal{L}_{g_3} + \lambda_4 \cdot \mathcal{L}_{g_4}$, where $\mathcal{L}_{g_i}(i = 1, 2, 3, 4)$ define the loss of our four subnetworks. Each of them could be described as

---

[6]Our training (test) images are generated from the training (test) images of the Foggy Cityscapes-DBF dataset [72].

Table 1: Quantitative evaluation results on synthetic data among our method, a representative polarization-based dehazing algorithm SPCVE [54] (taking three polarized images as the input), and five state-of-the-art learning-based dehazing methods including GDN [44], BPP [82], FFA [65], HardGAN [5], and MSBDN [7] (taking a single hazy image as the input).

|         | Ours  | SPCVE [54] | GDN [44] | BPP [82] | FFA [65] | HardGAN [5] | MSBDN [7] |
|---------|-------|------------|----------|----------|----------|-------------|-----------|
| PSNR    | **28.32** | 15.94  | 26.54    | 24.93    | 26.84    | 26.22       | 26.94     |
| MS-SSIM | **0.951** | 0.521  | 0.928    | 0.915    | 0.934    | 0.928       | 0.932     |

$\mathcal{L}_{g_i} = 2 \cdot \mathcal{L}_1 + \mathcal{L}_2$, where $\mathcal{L}_1$ and $\mathcal{L}_2$ denote the $L_1$ and $L_2$ loss respectively. $\lambda_i (i = 1, 2, 3, 4)$ are empirically set to be 1.0, 1.0, 2.0, 2.0 respectively.

**Training strategy.** We implement our network using PyTorch on an NVIDIA 2080Ti GPU and apply a two-phase training strategy. First, to ensure a stable initialization of the training process, we train our two network stages independently for 400 epochs. ADAM optimizer [26] is used with an initial learning rate $5 \times 10^{-4}$ for the first 300 epochs, and a linear decay to $2.5 \times 10^{-4}$ in the next 100 epochs. Then, we finetune the entire network in an end-to-end manner for another 300 epochs, keeping the learning rate to $5 \times 10^{-4}$. Instance normalization [90] are added during training.

## 5 Experiments

### 5.1 Evaluation on synthetic data

We compare our results to a representative polarization-based dehazing algorithm SPCVE [54] which also takes three polarized images as the input and five state-of-the-art learning-based dehazing methods including GDN [44], BPP [82], FFA [65], HardGAN [5], and MSBDN [7] which take a single hazy image as the input[7]. SPCVE [54] assumes that the transmitted light is not significantly polarized ($\mathbf{P}_T = \mathbf{0}$), and uses optimization to estimate $\mathbf{P}_A$ and $\mathbf{A}_\infty$, while our method takes into account the polarization effects of transmitted light and adopts deep learning to estimate $\mathbf{P}_T$, $\mathbf{P}_A$, and $\mathbf{A}_\infty$. All of these learning-based methods are re-trained on our dataset using $\mathbf{R}$ and $\hat{\mathbf{I}}$ (the calculated hazy image from $\mathbf{I}_{\alpha^{(i)}} (i = 1, 2, 3)$ using the linear system from Equation (8))[8]. Note that comparing with learning-based dehazing methods might be a bit unfair because of the difference in types of input data (ordinary image *vs.* polarized image), and we conduct such a comparison to show the advantage of using polarized images over image-only approaches.

Visual quality comparisons of dehazed results are shown in Figure 3[9]. Compared to the polarization-based dehazing algorithm SPCVE [54], our method can dehaze robustly with fewer artifacts; compared to the learning-based methods, our method performs better in recovering details. Taking the sky region (green box) in the first group of Figure 3 as an example, SPCVE [54] suffers severely from noise, and the learning-based methods yield bad pixels (shown as black streaks in the sky). This is because in our synthetic dataset we not only simulate the polarization effects of airlight but also transmitted light, and add spatially-variant turbulence to the scattering process, while SPCVE [54] ignores the polarization effects of the transmitted light and does not consider semantic and contextual information to refine the results, and the learning-based methods [44, 82, 65, 5, 7] are prone to artifacts for the pixels with large spatially-variant turbulence. To evaluate the results quantitatively, we adopt two frequently-used image quality metrics including PSNR and MS-SSIM (multi-scale SSIM). Results are shown in Table 1 (also below corresponding examples in Figure 3). Our model consistently outperforms the polarization-based and learning-based dehazing methods on all metrics.

---

[7]Note that the code of SPCVE [54] is not available and the demonstrated results are based on our own implementation. We directly provide the ground truth $\mathbf{A}_\infty$ to our implementation as its upper bound performance, also owing to that SPCVE [54] requires similar objects or known depth to estimate $\mathbf{A}_\infty$, which are not always available in our scenes.

[8]We should not use the ground truth $\mathbf{I}$ as the input of these learning-based methods since we could only get $\hat{\mathbf{I}}$ during the inferring phase of real data; and if we use $\mathbf{I}$ to re-train them, their results will be degenerated due to the large domain gap between $\mathbf{I}$ and $\hat{\mathbf{I}}$ (caused by the noise in $\mathbf{I}_{\alpha^{(i)}} (i = 1, 2, 3)$).

[9]Additional synthetic results can be found in the supplementary material.

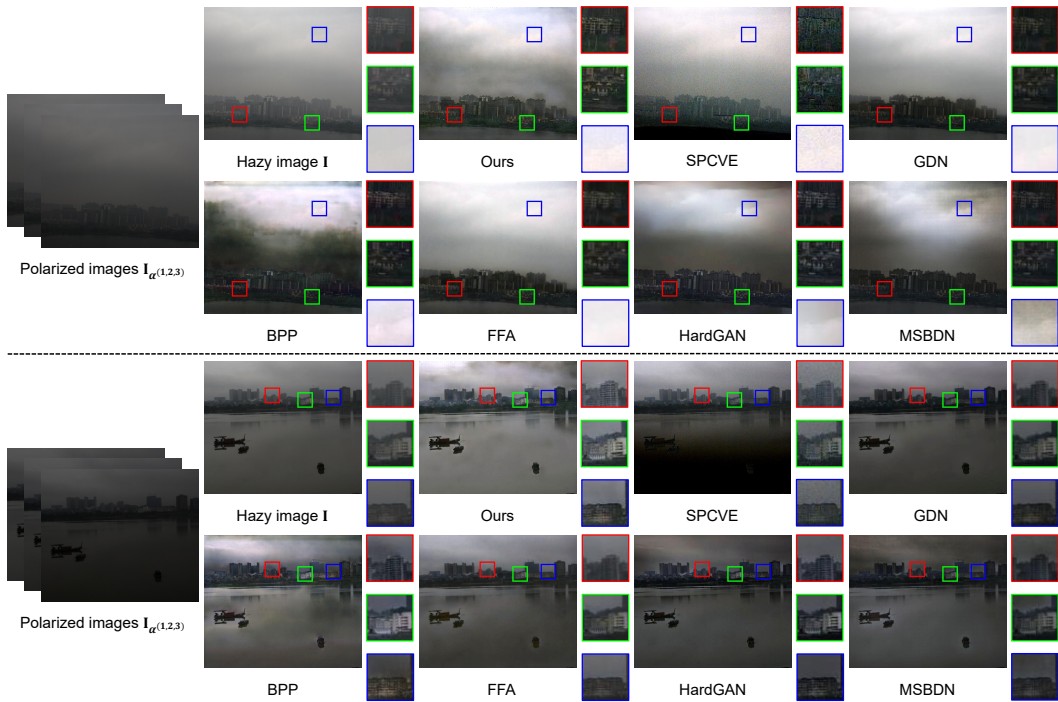

Figure 4: Qualitative comparisons on real data. See the caption of Figure 3 for explanation. All dehazing results are white-balanced to the similar color appearance and multiplied by a factor of 1.25 for better visualization. Please zoom-in for better details.

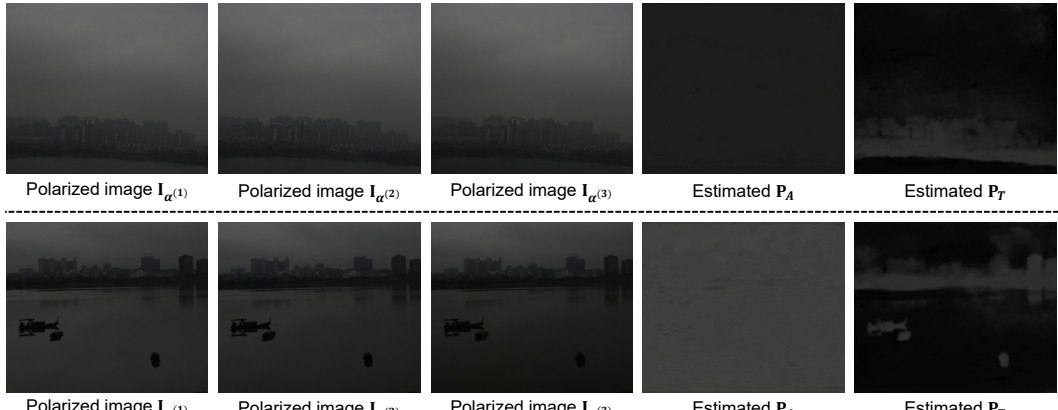

Figure 5: The polarized images side by side with the estimated $\mathbf{P}_A$ and $\mathbf{P}_T$ on real data.

## 5.2 Evaluation on real data

We use the Lucid Vision Phoenix polarization camera (RGB) to capture real data. The polarization camera can take four images with different polarization angles ($0°$, $45°$, $90°$, and $135°$) at a single shot. We use three of them ($0°$, $45°$, and $90°$) as the input to our method and SPCVE [54], and calculate the hazy image $\hat{\mathbf{I}}$ from the polarized images $\mathbf{I}_{\alpha^{(i)}}(i = 1, 2, 3)$ using the linear system from Equation (8) as the input to learning-based methods (GDN [44], BPP [82], FFA [65], HardGAN [5], and MSBDN [7]). Visual quality comparisons of dehazed results are shown in Figure 4[10]. Our method is able to generate clearer and brighter images than those by the state-of-the-art polarization-based and learning-based methods. For example, the color of the buildings (red box) in the first group of Figure 4, is correctly recovered by our method, while other methods suffer from color distortion

---

[10]Additional real results can be found in the supplementary material.

Table 2: Quantitative evaluation results of ablation study.

|  | PSNR | MS-SSIM |
|---|---|---|
| Ignoring the polarization effects of the transmitted light | 27.63 | 0.943 |
| Neglecting the spatially-variant real-world scattering | 27.86 | 0.948 |
| Directly estimating $\mathbf{T}$ and $\mathbf{R}$ | 27.27 | 0.945 |
| Removing $g_2$ ($g_4$) | 21.55 (21.28) | 0.740 (0.903) |
| Removing both $g_2$ and $g_4$ | 17.04 | 0.662 |
| Our final model | **28.32** | **0.951** |

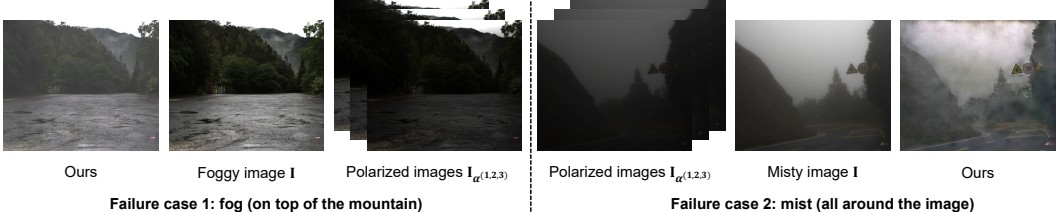

Ours    Foggy image $\mathbf{I}$    Polarized images $\mathbf{I}_{\alpha^{(1,2,3)}}$    Polarized images $\mathbf{I}_{\alpha^{(1,2,3)}}$    Misty image $\mathbf{I}$    Ours

**Failure case 1: fog (on top of the mountain)**    **Failure case 2: mist (all around the image)**

Figure 6: Two failure cases (fog and mist) in which our method shows degenerate performance.

artifacts which dim the results. For better visualization, we also show the polarized images side by side with the estimated $\mathbf{P}_A$ and $\mathbf{P}_T$ in Figure 5. We can see that the distributions of $\mathbf{P}_A$ and $\mathbf{P}_T$ satisfy the ones mentioned in Section 3.1, which demonstrates the rationality of our motivation.

### 5.3 Ablation study

To verify the validity of each model design choice, we conduct a series of ablation studies and show comparisons in Table 2. We first show the effectiveness of our physical image formation model by comparing with a model that ignores the polarization effects of the transmitted light (by taking $\mathbf{P}_T$ as zero) and a model that neglects the spatially-variant real-world scattering (by taking $\mathbf{P}_T$, $\mathbf{P}_A$, and $\mathbf{A}_\infty$ as spatially-uniform parameters). From the results we can see that our model is more generalized and reasonable. We further verify the contribution of our dehazing pipeline which estimates three parameters ($\mathbf{P}_T$, $\mathbf{P}_A$, and $\mathbf{A}_\infty$) to solve $\mathbf{T}$ and $\mathbf{R}$ by comparing with a model that directly estimates $\mathbf{T}$ and $\mathbf{R}$. We find that our dehazing pipeline is better than directly estimating $\mathbf{T}$ and $\mathbf{R}$ since these parameters are easier to learn than $\mathbf{T}$ and $\mathbf{R}$. Then, we demonstrate the necessity of the refinement subnetworks ($g_2$ and $g_4$) by removing $g_2$, $g_4$, and both of them. We could tell that without the refinement subnetworks[11], the performance degenerates rapidly, while it still outperforms the existing polarization-based (also optimization-based) dehazing algorithm SPCVE [54] (see Table 1 for the performance of SPCVE [54]) thanks to our generalized physical image formation model.

## 6 Conclusion

We presented a learning-based solution which leverages the properties of polarized light for image dehazing. To handle the images captured in the wild, we proposed a generalized physical formation model of hazy images, introduced a robust polarization-based dehazing pipeline, and designed a neural network tailored to the pipeline, showing state-of-the-art performance. Our solution extended the applicability of polarization-based dehazing methods by adopting deep learning to estimate the infinite airlight and the DoP of both transmitted light and airlight without the requirement of specific clues (*e.g.*, sky regions, similar objects), while considering the spatially-variant real-world scattering.

**Limitations.** Since our method is based on the physical image formation model of hazy images, it may fail in situations which does not conform to the model, such as fog or mist. As shown in Figure 6, our method shows degenerate performance on those images, because fog and mist are caused by a suspension of water droplets, while haze is a suspension of extremely small particles (other than water droplets) in the air. As future work, we plan to extend our model to support other situations.

---

[11]Synthetic results without refinement can be found in the supplementary material.

## Acknowledgments and Disclosure of Funding

This work is supported by National Key R&D Program of China (2020AAA0105200), and National Natural Science Foundation of China under Grant No. 62136001, 62088102, 61872012, 61876007.

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
