# Supplementary Material:
# Learning to dehaze with polarization

**Chu Zhou**[1] **Minggui Teng**[2] **Yufei Han**[5] **Chao Xu**[1] **Boxin Shi**[2,3,4]*

[1]Key Lab of Machine Perception (MOE), Dept. of Machine Intelligence, Peking University
[2]Nat'l Eng. Lab for Video Technology, School of Computer Science, Peking University
[3]Institute for Artificial Intelligence, Peking University
[4]Beijing Academy of Artificial Intelligence
[5]School of Info. and Comm. Eng., Beijing University of Posts and Telecommunications
{zhou_chu, minggui_teng, shiboxin}@pku.edu.cn,
hanyufei@bupt.edu.cn, xuchao@cis.pku.edu.cn

## 7 Additional synthetic results

In this section, we provide additional comparisons on synthetic data among our method, a representative polarization-based dehazing algorithm SPCVE [4] which also takes three polarized images as the input, and five state-of-the-art learning-based dehazing methods including GDN [3], BPP [6], FFA [5], HardGAN [1], and MSBDN [2] which take a single hazy image as the input, as shown in Figure 7, Figure 8, and Figure 9, corresponding to Footnote 9 in Section 5.1 of the paper.

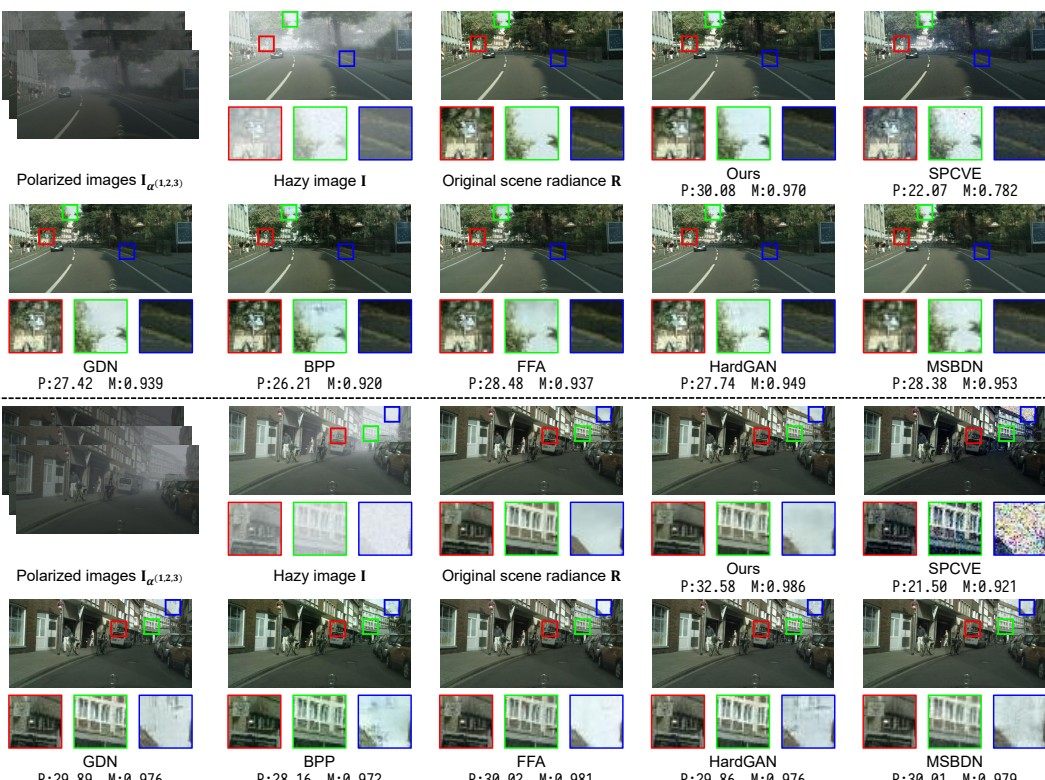

Figure 7: Additional comparisons on synthetic data (part 1). Quantitative results evaluated using PSNR (P) and MS-SSIM (M) are displayed below each image.

---

*Corresponding author.

35th Conference on Neural Information Processing Systems (NeurIPS 2021).

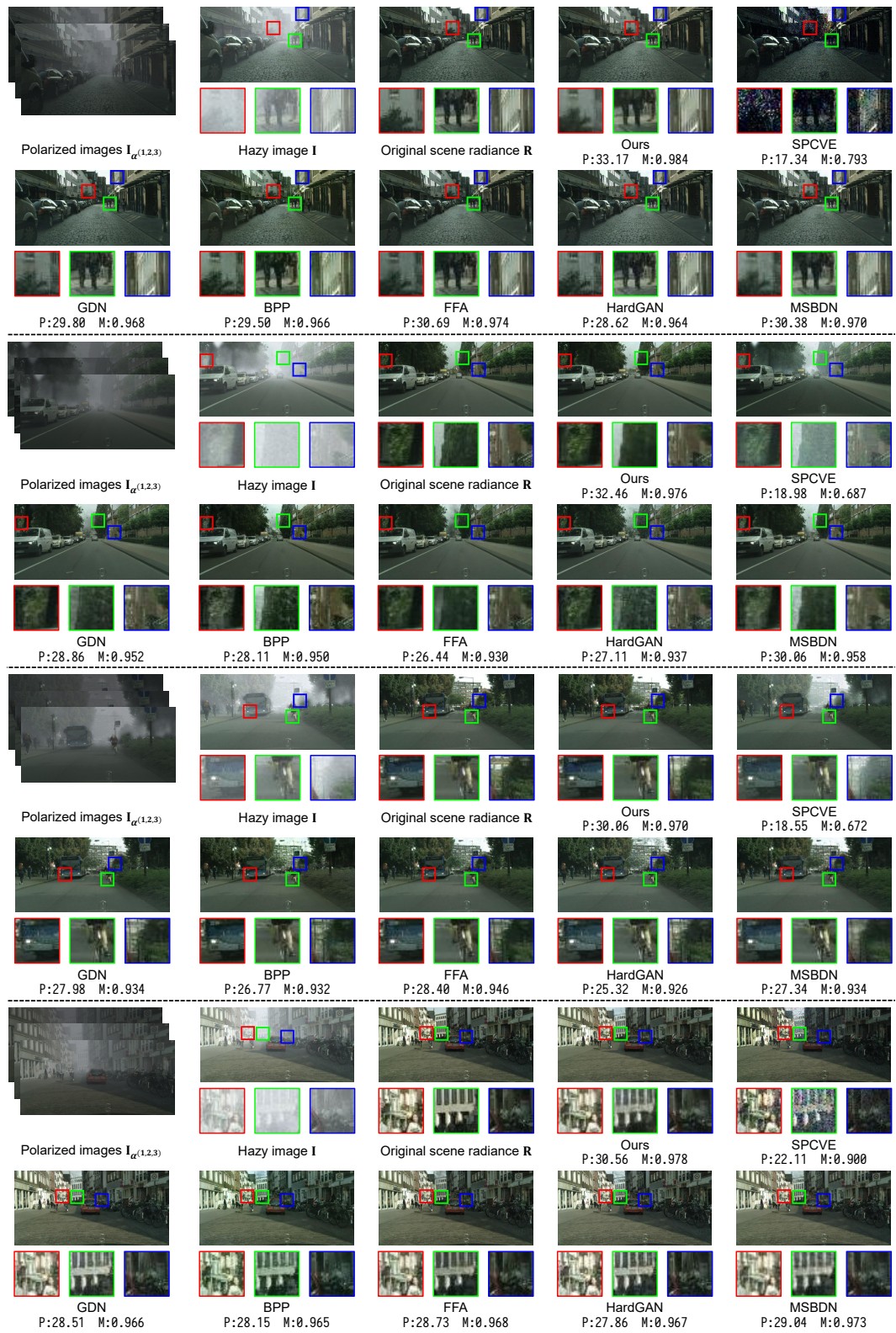

Figure 8: Additional comparisons on synthetic data (part 2). Quantitative results evaluated using PSNR (P) and MS-SSIM (M) are displayed below each image.

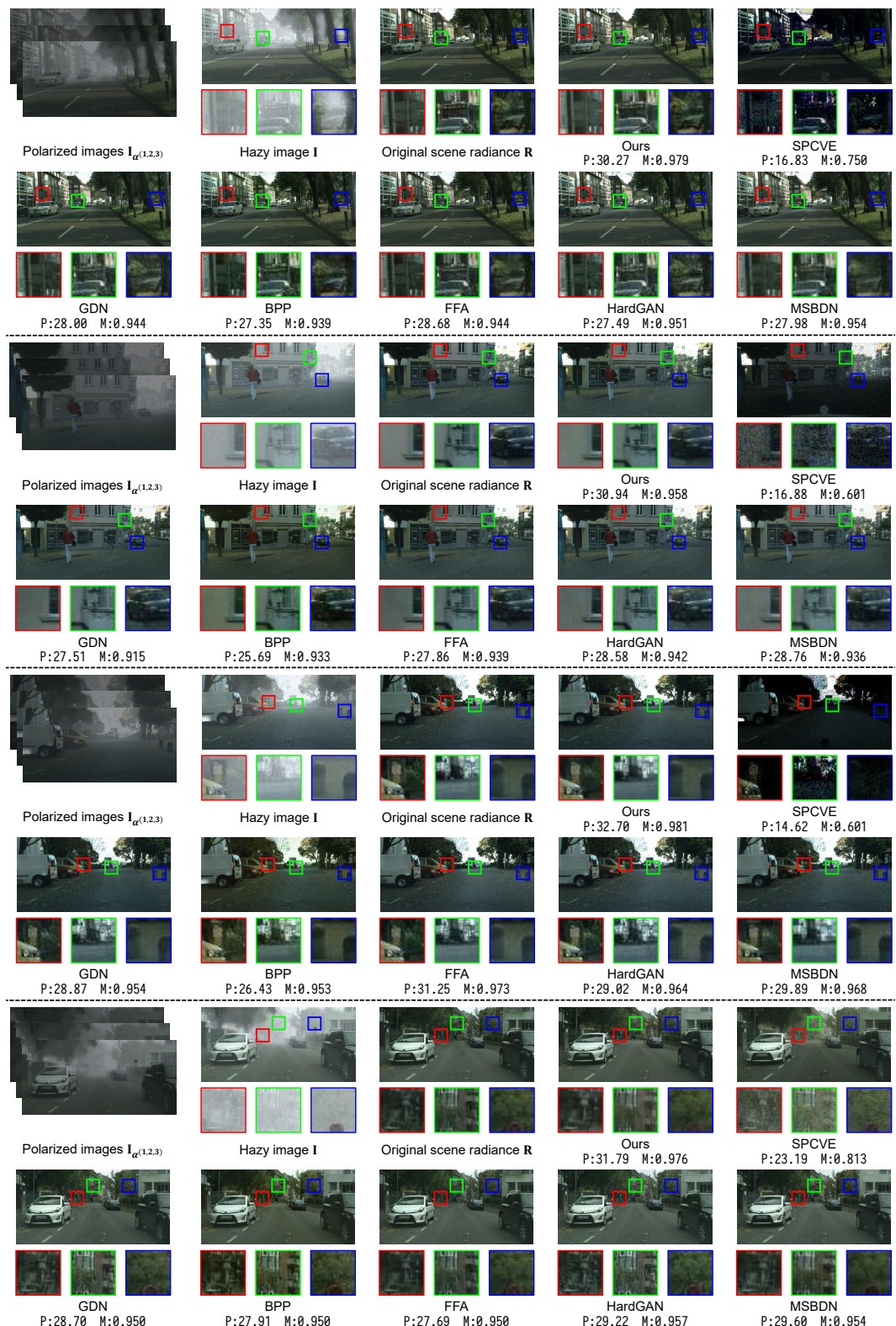

Figure 9: Additional comparisons on synthetic data (part 3). Quantitative results evaluated using PSNR (P) and MS-SSIM (M) are displayed below each image.

# 8 Additional real results

In this section, we provide additional qualitative comparisons on real data among our method, a representative polarization-based dehazing algorithm SPCVE [4] which also takes three polarized images as the input, and five state-of-the-art learning-based dehazing methods including GDN [3], BPP [6], FFA [5], HardGAN [1], and MSBDN [2] which take a single hazy image as the input, as shown in Figure 10, corresponding to Footnote 10 in Section 5.2 of the paper.

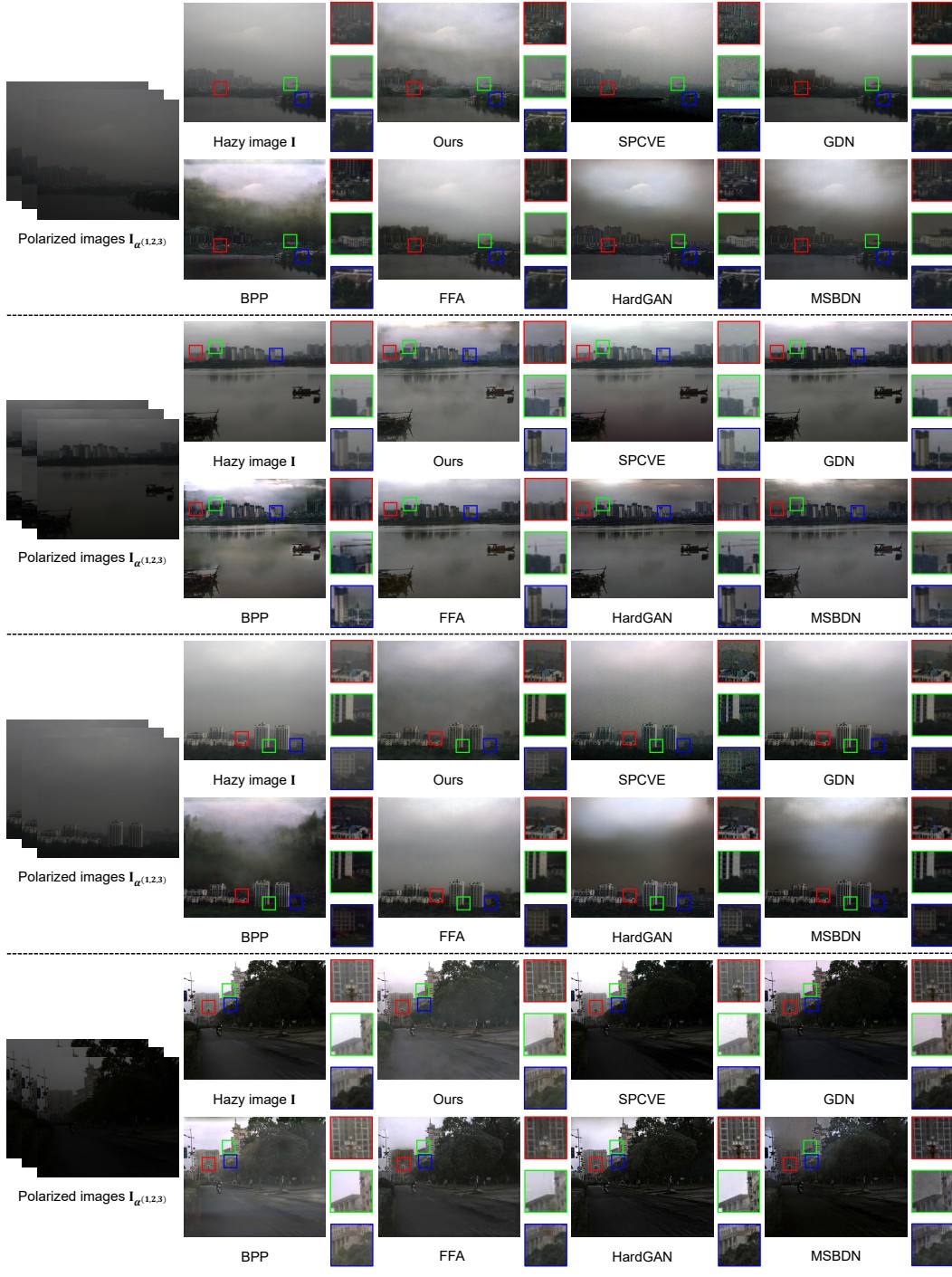

Figure 10: Additional qualitative comparisons on real data. All dehazing results are white-balanced to the similar color appearance and multiplied by a factor of 1.25 for better visualization.

# 9 Synthetic results without refinement

In this section, we provide qualitative comparisons on synthetic data without refinement (including the transmitted light **T** and the original scene radiance **R**), as shown in Figure 11, corresponding to Footnote 11 in Section 5.3 of the paper.

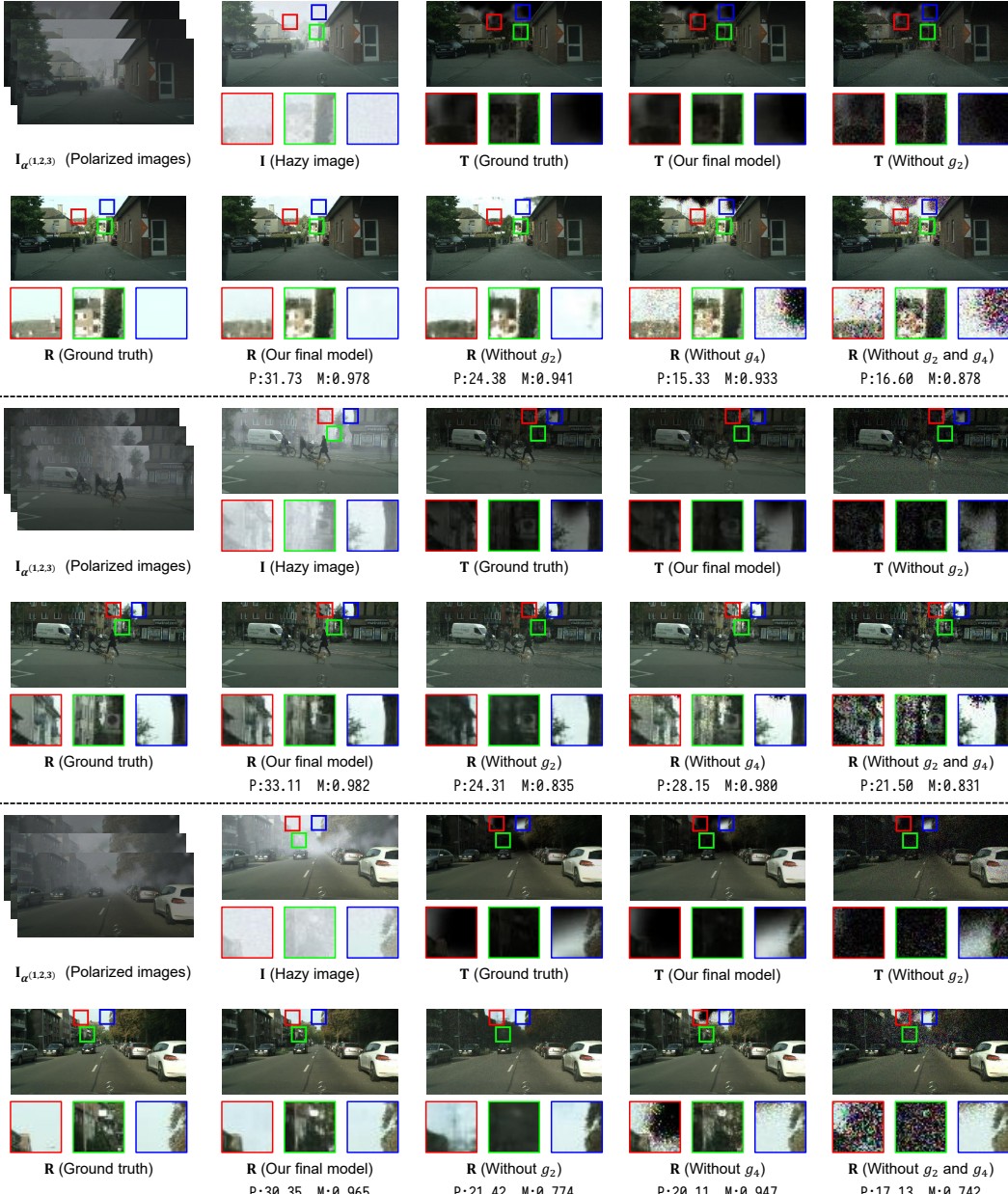

Figure 11: Qualitative comparisons on synthetic data without refinement (including the transmitted light **T** and the original scene radiance **R**). Quantitative results evaluated using PSNR (P) and MS-SSIM (M) are displayed below each image.