# OpenReview forum: "Learning to dehaze with polarization"
_NeurIPS.cc/2021/Conference — NeurIPS 2021 Poster_

### Official Review · Reviewer_bb8S · 2021-07-15

**Rating:** 7
**Confidence:** 4

**Summary:**

The paper presents a method for polarization-based dehazing of images. Using a camera with a polarization filter array, multiple polarization images can be captured in a single snapshot. These images are used to estimate an unpolarized image $I$ and a degree-of-polarization image $P$. A series of neural networks are trained on synthetic data to estimate the polarization and radiance of the ambient and transmitted scene light conditioned on $I$ and $P$. The results of the dehazed estimated scene radiance are compared with other methods for dehazing using polarization images or a single unpolarized image. Quantitatively, the approach appears to offer roughly 1-2 dB in improvement over previous methods in terms of PSNR. Qualitatively, the method appears to generate improved results on captured data as well.



**Limitations And Societal Impact:**

Limitations and societal impact appear to be adequately addressed.

**Main Review:**

Overall, the paper seems to be well written and the method is clearly explained. The results convincingly demonstrate that accounting for polarization in the dehazing task can improve image quality. While the results do not offer an overwhelming improvement over previous techniques for single image dehazing, the improvement is significant. I think the method could inspire future work on polarization-based dehazing and inverse scattering methods.

I have just a few comments which I note below.

I was surprised by the poor performance of the SPCVE method. In their original paper the method appears to offer a good deal of qualitative improvement over the original images whereas the results shown in Figs. 3 and 4 for SPCVE are almost worse than the input images in some cases. What are the main reasons for this difference? Are there situations in which SPCVE is more competitive with the proposed approach?

Part of the motivation for the method is that regions with the same material properties often reflect light with similar polarization properties. I think seeing these properties in captured data would help motivate the method. For example, I would be curious to see the polarization images side by side (rather than in the stacked format of Figure 3, which obscures their differences). I'm also curious to see the intermediate results of the network, such as the predicted degrees of polarization (these could perhaps be included in the supplemental).

How are the simulation parameters described in section 4.1 chosen?

How do the predicted degree of polarization images in the captured data compare to the simulated dataset?

Fig. 2 typo \hat{T}->R after Eq. (7)(b)


**Time Spent Reviewing:**

2

---

> ### Author Response · Authors · 2021-08-08
> **For reviewer bb8S**
>
> * About the poor results of SPCVE [54] in our experiments.
>
>   * Our synthetic dataset not only simulates the polarization effects of airlight but also transmitted light, and adds spatially-variant turbulence to those physical quantities (*e.g.*, DoP), while SPCVE [54] is not good at handling these data (detailed reasons could be found at "Why the results of SPCVE [54] tend to be noisy." in the **common issues**).
>   * Our real images were captured in a relatively more inhomogeneous atmosphere than their original images shown in their paper (which could be downloaded at [their project homepage](https://webee.technion.ac.il/people/yoav/research/instant-dehazing.html)). And their original images are also suffer from noise artifacts.
>   * There could be situations where SPCVE [54] performs better, such as scenes with a relatively homogeneous atmosphere and not significantly polarized transmitted light. Besides, reducing the noise level of the input polarized images could increase its performance.
>
> * Polarization images side by side and intermediate results of the network such as DoP in captured data.
>
>   * They can be found at [this link](https://0x0.st/-46h.png), where $\textbf{P}_A$ and $\textbf{P}_T$ denote the DoP of airlight and transmitted light respectively. The assumption "regions with the same material properties often reflect light with similar polarization properties" is reasonable and intuitive for image dehazing problems because haze increases with distance, and the polarization properties of the light reflected by objects in the distance is less affected by other factors (such as surface normal).
>
> * How to chose the simulation parameters in synthetic dataset generation pipeline and how do the predicted DoPs in the captured data compare to the simulated dataset.
>
>   * The range of the scattering coefficient are from [32] with some adjustment (changing the sampling space from discrete to continuous), and the range of the polarization-related parameters (*e.g.*, $\textbf{P}_A$ in $[0.05,0.4]$ and $\textbf{P}_T$ in $[0.025,0.2]$, as shown in Section 4.1) are from the statistics in [13]. And the predicted DoPs ($\textbf{P}_A$ and $\textbf{P}_T$) in the captured data are shown in [this link](https://0x0.st/-46h.png). We can see that their distributions are similar to the synthetic ones, which demonstrates that our synthetic dataset generation pipeline is reasonable.

---

> > ### Comment · Reviewer_bb8S · 2021-08-28
> > **Response**
> >
> > I'd like to thank the authors for their response, which has helped clear up some questions I had during the review. After reading the other reviews, I would still keep my positive rating for the paper, and would encourage the authors to clarify the issues that were raised in the review (e.g., SPCVE, refinement modules...) in the camera ready if accepted.

---

### Official Review · Reviewer_N2H7 · 2021-07-16

**Rating:** 6
**Confidence:** 3

**Summary:**

This paper presents dehazing algorithm using characteristics of polarized images. It first introduces generalized physical formation model of polarized hazy images, regarding spatially-variant real-world scattering. Taking three differently polarized hazy images as input, the proposed algorithm estimates and refines degrees of polarization (DoP) of the transmitted light T and airlight A, and utilizes these predictions to finally obtain original scene radiance R. By doing so, this paper was able to show outstanding performances, especially on real-world images due to its consideration on spatially-variant DoP and global airlight.

**Ethical Concerns:**

.

**Limitations And Societal Impact:**

I appreciate the authors for mentioning the limitation regarding the generality of the proposed hazy image formation model, for other image degradation weather conditions.

**Main Review:**

The physical formulation and the dehazing pipeline that were proposed in this paper are reasonable, clearly described, and properly executed. The paper is well-written and was easy to follow. While I am satisfied overall, I have a few questions.

1. In refining process for both T and R, the authors claimed that the refinement was done by ‘using semantic and contextual information extracted from I(1, 2, 3)’. While removing these refining subnetworks does drop performance severely (table 2), it seems like the subnetworks are simply using I(1, 2, 3) as inputs and no other technical consideration using architecture or loss function was given. Please elaborate on why do the authors think that this kind of simple process can boost so much of a performance.

2. On top of the mention from the paper, ‘comparing with learning-based dehazing methods might be a bit unfair because of the difference in types of input data’, I think it also restricts practicality and applicability. The most commonly used image capture device is an RGB camera, and these ‘learning-based methods’ can be applied to any hazy image while the proposed algorithm cannot. I want to know the authors’ opinion on why the polarization-based dehazing approaches should be researched.

Minor comments: In figure 2, bottom block, initially estimated R is marked as T^.


**Time Spent Reviewing:**

10

---

> ### Author Response · Authors · 2021-08-08
> **For reviewer N2H7**
>
> * Why simple refinement subnetworks could increase the performance so much.
>   * Please see "The efficacy of refinement subnetworks ($g_2$ and $g_4$)" in the **common issues**.
>
> * Why the polarization-based dehazing approaches should be researched.
>   * Although the most commonly used image capture devices are RGB cameras, recently unconventional cameras are widely used to improve the experience of digital photography. For example, the latest smartphones (such as HUAWEI P40 Pro) are equipped with a variety of unconventional cameras (*e.g.*, ToF camera and color temperature sensor) to work along with RGB ones for better image quality. The unconventional sensors capture extra scene information beyond RGB, which would be helpful in conquering bottleneck issues of RGB-only image enhancement. Polarimetric image capture could also provide complementary information to RGB. Polarization-based dehazing methods [77, 78, 53, 81, 76, 25, 54, 50, 41, 40, 39, 67, 80, 13, 29] have been researched during these years, however, none of them adopts the generalized image formation model as ours without relying on specific clues, which limits their applicability. We believe it is a meaningful trial to combine deep-learning and polarization information, and it could inspire future work on polarization-based dehazing and inverse scattering methods (Reviewer bb8S). Besides, with the development of polarization cameras, capturing multiple images with different polarization angles in a single shot becomes available, which makes our approach potentially practical.

---

### Official Review · Reviewer_WrNr · 2021-07-18

**Rating:** 6
**Confidence:** 3

**Summary:**

This paper proposes a neural network for image dehazing where the input is
expected to contain polarization information (e.g. multiple polarized images of
the exact same scene). The total irradiance and total degree of polarization
can be obtained directly from the polarized input frames. The paper then
follows the image formation model of [77] (and others) to identify key
quantities to estimate: the infinite airlight color, and the degree of
polarization of the ambient and scene light. Multiple neural networks
are designed to estimate these components; they are coupled using
closed-formed expressions derived from the image formation model to produce the
final dehazed image. The networks are trained independently, then finetuned
jointly on a synthetic dataset.


**Ethical Concerns:**

None that struck me.

**Limitations And Societal Impact:**

The paper shows and discuss the limitations of their work.

**Main Review:**

This paper is well-written, consistent and produces good result. Despite its
claims, there is not much novelty in the image formation model. Like many
'physically-based' deep learning image restoration approaches, the proposed
technique replaces heuristic measurements of physical quantities (e.g. a patch
of sky to estimate the airlight) by neural network estimators.


### Strengths

+ I found the paper clear and easy to follow.
+ The description of prior art appears complete and relevant.
+ The derivation is sound (although not technically a contribution of this
work).
+ Synthetic test results are convincing and outperform previous single- and
multi- frame baselines.


### Weaknesses

- The complete derivation of section 3.1 is from [77], this should be better acknowledge.
- One the claimed contributions is a generalized hazy image formation model,
but it is not clear how this paper generalizes the ideas of [77]. The spatially
varying noise terms N(x, c) are introduced but never used in the derivation.
- The real-world results (Fig4) are not convincing. The image suffers from severe
grid artifacts, corrupted boundaries, ghosting around buildings. This makes
me question the "robustness" property claimed in the intro (2), it is not evident
at all that synthetic training data + the physical models are sufficient to
make the method practical in the real-world.
- How do the result degrade visually without the "refinement" networks? It
would be helpful to have some extra real/synthetic images in supplemental.


### Misc

- 268: dehzaing -> dehazing
- 28: although [...] however -> one suffices
- 41: for enabling -> to enable
- paragraph l.41 and l.53 of the intro read as a bit redundant. Maybe a contribution list is not  needed,
or the previous paragraph can be shortened?
- 123: divide -> split, decompose
- 134: consider using different notations for the spatial variation terms N(x, c).
I found it confusing to see it appear in theta, beta and A_\inf
- 201: Is Gaussian noise really a good model for the beta and alpha_inf
spatial variations? This choice seems a bit ad hoc
- The results from SPCVE seem very noisy


**Time Spent Reviewing:**

2

---

> ### Author Response · Authors · 2021-08-08
> **For reviewer WrNr**
>
> *  Novelty of the image formation model and the restoration approaches.
>    * As shown in Section 3.1, our physical image formation model is a generalized version of [77],  taking into account the polarization effects of both transmitted light and airlight (see Equation (2), (3), (5), (6), (7)), along with the spatially-variant real-world scattering (see the definition of those physical quantities in Line 119, 134). In [77], like other polarization-based dehazing methods [78, 53, 81, 76, 25, 54, 50, 41, 40, 39, 67, 80], they only consider the polarization effects of airlight (see Equation (2), (6), (8) in [77]) and assume that the DoP and infinite airlight are spatially-invariant (see Section 7 in [77]).
>    * And by adopting learning-based approaches, our method does not require specific clues (*e.g.*, sky regions) to estimate the DoP and infinite airlight, which extends the applicability. As far as we know, there is no such a polarization-based dehazing method that adopts the generalized image formation model as ours without relying on specific clues so far.
> *  About the spatially-variant turbulence $N(x,c)$.
>    * In our derivation, all operations are pixel-wised, and we always use **bold** letters to denote the spatially-variant physical quantities. Taking the infinite airlight $\textbf{A}_\infty=\{\bar A_\infty(c)+N(\textbf{x},c)\}$ for example, $\textbf{A}_\infty$ could be regarded as the sum of the mean value $\bar A_\infty(c)$ (spatially-invariant) and the turbulence $N(x,c)$ (spatially-variant), which means that $\textbf{A}_\infty$ already encodes the spatially-variant property. We will clearly define this in the final version.
> *  About real-world results.
>    * Although our method may lead to some ghosting artifacts around the buildings, it can generate images with less noise compared to the state-of-the-art polarization-based dehazing method SPCVE [54]. Besides, our results suffer less from color distortion artifacts than those by state-of-the-art learning-based methods [44, 82, 65, 5, 7] (as shown in Figure 4, our results are clearer and brighter). And we believe it is a meaningful trial to combine deep-learning and polarization information, and it could inspire future work on polarization-based dehazing and inverse scattering methods (Reviewer bb8S).
> *  The results without the refinement subnetworks and why they degrade visually.
>    * Please see "The efficacy of refinement subnetworks ($g_2$ and $g_4$)" in the **common issues**.
> *  Why we use Gaussian noise to simulate the spatially-variant property.
>    * Some polarization-based dehazing methods [13, 77] assume the atmosphere to be homogeneous, however, real atmosphere is inhomogeneous. As shown in the statistics in [39] and [41], the distributions of AoP (angle of polarization) and DoP (degree of polarization) approximately form a "bell-like" curve, and we empirically find that "mean value + Gaussian noise" is sufficiently accurate to fit such a curve.
> *  Why the results of SPCVE [54] tend to be noisy.
>    *  Please see "Why the results of SPCVE [54] tend to be noisy." in the **common issues**.

---

> > ### Comment · Reviewer_WrNr · 2021-09-10
> > **Thank you**
> >
> > Thank you for your detailed response. I have taken it into account for my final recommendation.

---

### Official Review · Reviewer_RnLz · 2021-07-19

**Rating:** 6
**Confidence:** 3

**Summary:**

This paper proposes a method for image dehazing by polarization-based learning pipeline. It is based on a physical formation model which is considering the spatially variant real-world scattering and taking into account the polarization effects of both transmitted light and airlight. Experiments demonstrate that the proposed method can dehaze robustly with fewer artifacts, and performs better in recovering details.


**Limitations And Societal Impact:**

The author fully addresses the physical image formation model of hazy images, and mentions that it does not apply to such as fog or mist. The use of polarization information as a feature will have a significant impact as polarization cameras become more widely used.

**Main Review:**

It is a very interesting approach to utilize polarization information of both transmitted light and airlight as a feature for image dehazing. By introducing spatially-variant turbulence N(x,c) from polarization information, the effect of light scattering in the air can be well reflected spatially, and the dehazing results seem to be better. Although the input image data is different, the dehazing results from the proposed method are better than the results from other comparison methods and I believe that the proposed method is able to utilize the polarization information.

On the other hand, I am afraid that there may be some assumptions in the scene where the proposed method can be applied as the scenes used in the evaluation experiments are similar. I think polarization information can be very useful, but it can also lead to erroneous results due to noise effects. (e.g. the haze in some pixels is very thick and leaves little information of the transmitted light.) It would be better to conduct evaluation experiments of the proposed method for various scenes (depth, normals, colors). I feel that the efficacy of refinement subnetworks (g2 and g4) as well as some networks for polarization information is significant, so it would be good to do more evaluation of refinement subnetworks.

The paper is well-written and well-structured. There are only a few typos (e.g. ‘dehzaing ’ in Section 6 Conclusion) that can be addressed in another proofreading.


**Time Spent Reviewing:**

1.5 hour

---

> ### Author Response · Authors · 2021-08-08
> **For reviewer RnLz**
>
> * Evaluation for different scenes.
>   * We respectfully believe that the source dataset (The Foggy Cityscapes-DBF dataset [72], see Section 4.1) we used to generate our synthetic data is a large-scale dataset containing different scenes with different depths (in the range of $[0,\infty]$), normal, and color (it contains 30 different classes of objects). We think our synthetic dataset already contains various scenes, and we would like to add more diverse scenes to the dataset in our future work.
> * The efficacy of refinement subnetworks ($g_2$ and $g_4$).
>   * Please see "The efficacy of refinement subnetworks ($g_2$ and $g_4$)" in the **common issues**.

---

### Author Response · Authors · 2021-08-08
**Common issues**

We sincerely thank all reviewers for their valuable comments and suggestions. We will fix the typos and improve the figures as suggested in the final version. We feel encouraged that our physical image formation model and the learning-based dehazing pipeline are acknowledged by the reviewers:

* "It is a very interesting approach to utilize polarization information of both transmitted light and airlight as a feature for image dehazing." (Reviewer RnLz)
* "The physical formulation and the dehazing pipeline that were proposed in this paper are reasonable, clearly described, and properly executed." (Reviewer N2H7)
* "The results convincingly demonstrate that accounting for polarization in the dehazing task can improve image quality." (Reviewer bb8S)

And we also appreciate the reviewers for the acknowledgment of our writing quality:

* "The paper is well-written and well-structured." (Reviewer RnLz)
* "I found the paper clear and easy to follow. The description of prior art appears complete and relevant." (Reviewer WrNr)
* "The paper is well-written and was easy to follow." (Reviewer N2H7)
* "Overall, the paper seems to be well written and the method is clearly explained." (Reviewer bb8S)


We first address the common questions in the **common issues**, and answer each reviewer's specific questions in the corresponding comments.

- The efficacy of refinement subnetworks ($g_2$ and $g_4$). (Reviewer RnLz, WrNr, N2H7)

  - The results without refinement can be found at [this link](https://0x0.st/-46G.png). And if we remove the refinement subnetworks, several issues will occur to downgrade the performance: (1) the numerical problem will occur when the denominator of Equation (7) (a) approaches zero, which often happens in pixels where $\textbf{P}_T\approx\textbf{P}_A$ (the DoP of transmitted light and airlight are approximately the same); (2) the estimated $\textbf{P}_T$ and $\textbf{P}_A$ by $g_1$ are prone to be noisy which distorts the calculated $\widehat{\textbf{T}}$ (*e.g.*, "$\textbf{T}$ (Without $g_2$)"), because the spatially-variant turbulence is hard to learn due to its irregularities; (3) when the haze in some pixels is very thick and leaves little information of the transmitted light ($\textbf{T}\approx\textbf{0}$), the numerator of Equation (7) (b) approaches zero, which leads to a wrong result that $\textbf{R}\approx\textbf{0}$ (*e.g.*, "$\textbf{R}$ (Without $g_4$)"). From these results we can see that if we remove $g_2$, the pixels in $\textbf{T}$ will be very noisy and tend to be dark, which makes the pixels in $\textbf{R}$ noisier and darker; and if we remove $g_4$, the pixels with thick haze ($\textbf{T}\approx\textbf{0}$) would lead to a wrong result ($\textbf{R}\approx\textbf{0}$).

  - The purpose of these two subnetworks can be summarized as: (1) denoise the results and relieve the spatially-variant turbulence; (2) hallucinate the image contents which cannot be calculated by the physical model (*e.g.*, the pixels with thick haze leading to $\textbf{R}\approx\textbf{0}$) using semantic and contextual information.

  - Note that "physical model-based pipeline + simple post-processing modules" is a widely-used strategy in various image enhancement problems to further refine the results. For example, the experiment results show in Lyu *et.al.*[^1] are similar to ours: the refinement subnetworks could improve the visual quality of the final results. This is partly because the refinement task is not difficult to learn given majority of the problems have been solved using physical model-based constraints. Besides, even if we remove both $g_2$ and $g_4$, our method still outperforms the state-of-the-art polarization-based dehazing method SPCVE [54] (shown in Table 1 and Table 2), we think this could demonstrate the contribution of both the learning-based dehazing pipeline and the refinement subnetworks.

    [^1]: Youwei Lyu, Zhaopeng Cui, Si Li, Marc Pollefeys, and Boxin Shi. "Reflection separation using a pair of unpolarized and polarized images." *Advances in neural information processing systems* 32 (2019): 14559-14569.

* Why the results of SPCVE [54] tend to be noisy. (Reviewer WrNr, bb8S)
  * This is because: (1) they ignore the polarization effects of transmitted light, which leads to inaccurate estimation of the transmitted light $\textbf{T}$; (2) they do not consider the spatially-variant real-world scattering, and do not refine the reconstructed $\textbf{R}$; (3) the physical quantities (*e.g.*, DoP) estimated by specific clues without semantic and contextual information are prone to be erroneous (note that in the experiments of synthetic data, we only provide the ground truth infinite airlight, and DoP is estimated on their own).

---

### Decision · Program_Chairs · 2021-09-27

**Decision:**

Accept (Poster)

**Comment:**

The four reviewers thought this paper was above threshold for acceptance. They all found the idea useful and interesting. The author response also helped to clarify some issues raised by the reviewer.